

# Effect of fragmentation on the Costa Rican dry forest avifauna

Gilbert Barrantes,  Diego Ocampo,  José D. Ramírez-Fernández and  Eric J. Fuchs

Escuela de Biología, Universidad de Costa Rica, San José, Costa Rica

## ABSTRACT

Deforestation and changes in land use have reduced the tropical dry forest to isolated forest patches in northwestern Costa Rica. We examined the effect of patch area and length of the dry season on nestedness of the entire avian community, forest fragment assemblages, and species occupancy across fragments for the entire native avifauna, and for a subset of forest dependent species. Species richness was independent of both fragment area and distance between fragments. Similarity in bird community composition between patches was related to habitat structure; fragments with similar forest structure have more similar avian assemblages. Size of forest patches influenced nestedness of the bird community and species occupancy, but not nestedness of assemblages across patches in northwestern Costa Rican avifauna. Forest dependent species (species that require large tracts of mature forest) and assemblages of these species were nested within patches ordered by a gradient of seasonality, and only occupancy of species was nested by area of patches. Thus, forest patches with a shorter dry season include more forest dependent species.

## INTRODUCTION

Deforestation and change in land use are the primary factors causing habitat degradation and forest fragmentation in tropical regions (*Jaeger, 2000*; *Lambin, Geist & Lepers, 2003*; *Azevedo-Ramos, De Carvalho Jr & Do Amaral, 2006*; *Joyce, 2006*; *Martínez et al., 2009*). These changes in natural landscapes may often reduce connectivity for species trying to move between fragments embedded in a matrix that consists of anthropogenic and semi-natural habitats (*Renjifo, 2001*; *Graham & Blake, 2001*). Additionally, the negative effects of habitat fragmentation may be exacerbated by increased predation and competition within habitat patches (*Andrén, 1992*; *Fahrig, 2003*). Thus, the direct and indirect negative effects of habitat fragmentation could cause the extinction of some species in the fragments, particularly of those species that rely on large tracts of mature forest for reproduction (*Stiles, 1985*).

Bird communities undergo notable changes in composition and abundance soon after large forests are reduced into smaller patches (*Bierregaard & Stouffer, 1997*; *Oostra, Gomes & Nijman, 2008*). In some cases fragmentation could increase bird species richness and the abundance of bird species (*Azevedo-Ramos, De Carvalho Jr & Do Amaral, 2006*) as birds aggregate in the remaining available forests patches. But this increase is invariably followed

Corresponding author
Eric J. Fuchs, e.j.fuchs@gmail.com

by a steady reduction (*Herkert, 1994*; *Kruess & Tscharntke, 1994*; *Lida & Nakashizuka, 1995*), because the remaining habitat is insufficient to sustain either the increased abundance or a large number of species (*Fahrig, 2003*; *Pimm & Askins, 1995*). Additionally, invasion of non-forest bird-species may increase competition for resources and increase parasite load, which in turn may further reduce the reproductive success and viability of populations within fragments (*Christiansen & Pitter, 1997*; *Duncan, 1997*).

Reductions in genetic variability, demographic crashes and higher susceptibility to catastrophic events have been hypothesized for resident bird populations in forest patches (*Zuidema, Sayerand & Dijkman, 1996*). These effects may be prevented if fragments are interconnected or connected to larger continuous forests (*Haddad et al., 2003*; *Uezu, Metzger & Vielliard, 2005*). Unfortunately, due to species-specific differences in behavior, corridors are not suitable for all species (*Rosenberg, Noon & Meslow, 1997*). Furthermore, newly fragmented habitats are often difficult to re-connect with larger forest fragments or with other small fragments; in most cases continuous habitats are impossible to recover. Hence, isolated forest patches often become the only species reservoirs of previously widespread avifaunas.

Northwestern Costa Rican dry-forest has been reduced to a series of small patches surrounded by large cultivated areas (e.g., sugar cane, rice field, cattle haciendas; *Quesada & Stoner, 2004*) which cover only 0.1% of its original extension (*Janzen, 1988*). Consequently, the original terrestrial avifauna is now confined to these isolated, small forest patches; some of which are protected but with little, if any, chance of reconnection. As a first approach to estimate the importance of these forest fragments for the dry forest avifauna, we tested the effect of natural habitat fragmentation (*Fahrig, 2003*; *Fahrig, 2013*) on species composition in dry forests of northwestern Costa Rica at two different levels: the entire bird community, and the assemblage of forest dependent species. Most dry forest birds occupied originally nearly the entire northwestern region of Costa Rica, and even species that now occur at middle elevations were reported at lower elevations (*Wetmore, 1944*; *Slud, 1980*). Thus, it is likely that the distribution of most species has been affected by fragmentation due to habitat loss. We use nestedness analyses to test whether fragmentation or the length of the dry season (i.e., seasonality gradient) produce a nested species pattern. Testing if forest patches nest along a seasonal gradient provides information on the potential effect of climate changes predicted for the region (*Sheffield & Wood, 2008*). A nested pattern is expected when species assemblages in species-poor sites are a subset of those assemblages present in species-rich sites (patch nestedness), or when species occupying few sites are a subset of those species occupying a large number of sites (species nestedness, *Novak, Moore & Leidy, 2011*). Hence nestedness may be the result of variation in rates of colonization and extinction among sites (*Lomolino, 1996*), or among species (*Atmar & Patterson, 1993*). We also test whether extinction (the likely cause of species richness reduction across forest fragments) is caused by either a reduction in patch size, an increase in the distance between forest fragments as consequence of habitat deterioration, or if the nested pattern is associated with a climatic gradient (i.e., seasonality) (*Lomolino, 1996*; *Wright et al., 1998*).

**Table 1  Area, location and number of native species recorded in five dry forest fragments in northwestern Costa Rica.** Dry season includes the length of dry season in months and is based on meteorological stations located in the same or nearby sites.

| Locality | Dry season | Area (ha) | Location | No. of species |
|---|---|---|---|---|
| Santa Rosa | 6–6.5 | 37,117 | 10°50′N, 85°37′W | 123 |
| Palo Verde | 5–5.5 | 11,970 | 10°20′N, 85°20′W | 135 |
| Rincón de laVieja | 2–3 | 8,411 | 10°49′N, 85°21′W | 127 |
| Diriá | 4.5–5 | 5,426 | 10°10′N, 85°35′W | 109 |
| Cabo Blanco | 3–4 | 1,172 | 09°33′N, 85°06′W | 104 |

## MATERIALS AND METHODS

### Data collection

We gathered information on species composition from five forest fragments in northwestern Costa Rica that varied in size and connectivity (Table 1 and Fig. 1): Parque Nacional Diriá (Dir), Reserva Natural Absoluta Cabo Blanco (CB), Parque Nacional Palo Verde and the Reserva Biológica Lomas Barbudal (PV), Parque Nacional Santa Rosa (SR), and Parque Nacional Rincón de la Vieja (RV). These forest patches are surrounded by large agricultural fields and human communities (*Joyce, 2006*) and the distance between the nearest two forest patches included in this study is 58.9 km. There are some isolated trees and small tracts of early successional vegetation in the matrix that surround forest patches, but they are likely inadequate as corridors between patches. The fragmentation in the region has been the result of a progressive loss of natural habitats due to transformation of these habitats into agricultural fields (*Boucher et al., 1983*; *Joyce, 2006*). We visited (GB, DO and JDRF) each site from 8 to 20 times to compile a comprehensive bird list of each site during the last 15 years, starting in 2002. We sampled 3–4 days during each visit and searched for birds from 8 to 12 walking hours/day. In sites in which access is difficult (e.g., RV) we extended the sampling period of each visit to 5–8 days, to reduce the number of visits. Particularly during the breeding period of most dry forest birds (May through July) we focused our efforts in detecting those elusive, rare species (e.g., some nightjars and cuckoos). We complemented our survey data with information from *Stiles (1983)*, checklist of SR and OTS (PV); Julio Sánchez and Luis Sandoval provided us additional data for PV and RV respectively.

Climatically the northwestern region of Costa Rica is characterized by a long dry season from December through May (*Mata & Echeverría, 2004*) followed by a rainy season. However the local conditions affect the length of the dry season across sites (*Sánchez-Murillo et al., 2013*), and this makes it possible to order sites along a gradient of seasonality (Table 1). Precipitation patterns influence vegetation in the region, which is dominated by deciduous vegetation with evergreen species along rivers, and seasonal and permanent streams (*Hartshorn, 1983*). We obtained the area of each patch, for statistical analyses, from the Sistema Nacional de Areas de Conservación de Costa Rica (http://www.sinac.go.cr). For PV we excluded the area covered by wetlands and included the area of the Reserva Biológica Lomas Barbudal because it is connected with PV, and excluded the area on the Caribbean slope of RV because this area is covered with rain forests rather than dry forests (*Hartshorn, 1983*).

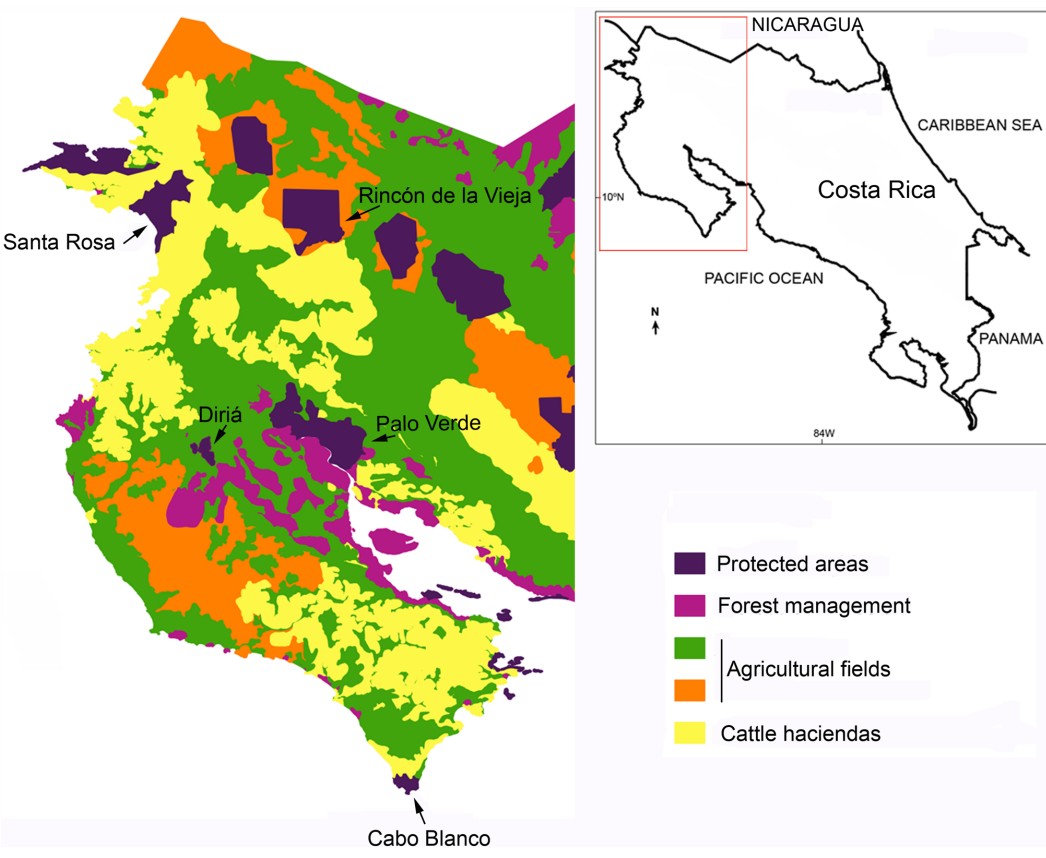

**Figure 1** **Dry forest fragments and land use in northwestern Costa Rica.** The inset shows the map of Costa Rica.

We excluded fresh water (e.g., Anatidae, Ardeidae) and marine birds (e.g., Fregatidae) from our analyses because they are restricted to habitats that are not present in all sites. We also excluded migratory birds. Occasional native species far from their normal geographic breeding distribution ranges were also excluded. Thus, the dry-forest bird-community was composed of resident, breeding species with terrestrial habits. Each bird species was classified into one of three forest dependency categories following *Stiles (1985)* with modifications by *Sandoval & Barrantes (2009)*: 1 = species that live and reproduce in extensive mature forest; 2 = species that require habitats with 40–50% of forest cover; 3 = species that inhabit open areas. Species in categories 1 and 2 are likely more affected by habitat fragmentation than those species included in category 3.

## Statistical analyses

We evaluated similarity of bird species composition among sites using Sørensen Index and tested for a relationship between species composition and geographic distances between sites using a Mantel test. We also determined whether composition of resident species across dry forest patches follows a nested distribution, or if each patch contains an independent subset of species, using the Vegan package (version 1.17; http://cran.r-project.org) implemented in the R Statistical Language (version 3.00; *R Core Team, 2013*). A nested species distribution

occurs when species richness in smaller fragments is a subset of the richness of larger fragments. To test for species nestedness in the dry forest patches we used presence/absence matrices in which rows and columns were patches and species respectively, and then sorted rows (assemblages) and columns (species) by a gradient of patch sizes, and a gradient of rainfall seasonality. We calculated the NODF metric (*Almeida-Neto et al., 2008*; *Ulrich, Almeida-Neto & Gotelli, 2009*; *Moreira & Maltchik, 2012*) and used 999 permutations with the 'quasiswap' algorithm to construct a null distribution of NODF values (*Miklós & Podani, 2004*). During each permutation the 'quasiswap' algorithm randomly shuffles values of rows and columns but maintains constant marginal frequencies (total frequencies of rows and columns). We then estimated the probability that the calculated nestedness differed significantly from the generated null distribution. 'Quasiswap' in addition to retaining row (sites) and column (species) frequencies, does not increase Type I or Type II errors (*Gainsbury & Colli, 2003*). We used the NODF metric because it independently estimates nestedness of species assemblages among sites (NODF rows), and nestedness for occupancy or presence among species (NODF columns), and for the entire matrix that we refer to as community (*Almeida-Neto et al., 2008*). NODF, for instance, calculates the nestedness of sites by comparing the occurrence of each species in each site (i.e., fill or empty cells) with the marginal values corresponding to all sites, and then ranking the sites by a previous determined gradient (e.g., area of fragments) (*Almeida-Neto et al., 2008*). NODF is reported in values ranging from 0 (not nested) to 100 (maximum nestedness). These models do not incorporate the probability of detection of each species, but require that all (or nearly so) species from each fragment are included.

We also inferred whether nestedness was either caused by extinction due to a reduction in patch size or by rainfall seasonality (*Cutler, 1991*; *Lomolino, 1996*; *Patterson, 1990*; *Patterson & Atmar, 1986*) conducting the statistical analyses described by *Lomolino (1996)*: %PN $= 100 \times (R - D)/R$, where %PN $= \%$ of perfect nestedness, $R =$ mean number of departures from random simulations, and $D =$ number of species that depart from perfect nestedness. To estimate the probability associated with the $D$ statistic, we took the ratio of species that depart from perfect nestedness, between the original matrix and those obtained from 999 randomly generated matrices (scripts for running these analyses are included as Supplemental Information 1). Lomolino's statistics were calculated for a presence/absence species matrix in which sites were first ordered by decreasing area and then along a gradient of seasonality (Fig. 1). Analyses were conducted for all the species and for forest dependent species (categories 1 and 2 of forest dependence). With these matrices we tested if nestedness is caused by extinction due to area reduction or rainfall seasonality.

## RESULTS

We registered a total of 187 resident species in all study sites (Table S1). PV had the most resident species while CB and Dir had the fewest, but the number of species did not differ significantly across sites ($x^2 = 5.51$, $df = 4$, $p = 0.239$; Table 1). RV was least similar in species composition with all other sites, and CB and Dir were most similar (Table 2). Species richness was independent of both geographic distance (Mantel test $= 0.42$, $p = 0.185$; Table 2) and fragment area ($r = 0.41$, $p = 0.494$).

**Table 2** **Sørensen similarity index and the number of species shared between sites in parentheses (below the diagonal) and distance (km) (above the diagonal) between five forest patches in northwestern Costa Rica.** Larger values of the Sørensen similarity index indicate greater similarity in species composition between sites.

|  |  | Distance | | | | |
| --- | --- | --- | --- | --- | --- | --- |
|  |  | S. Rosa | P. Verde | Diria | C. Blanco | R. Vieja |
|  | S. Rosa |  | 77.9 | 72.6 | 71.5 | 63.6 |
|  | P. Verde | 0.80 (116) |  | 68.9 | 69.4 | 58.9 |
| Similarity | Diria | 0.82 (92) | 0.71 (97) |  | 70.6 | 61.3 |
|  | C. Blanco | 0.81 (88) | 0.67 (94) | 0.85 (86) |  | 64.6 |
|  | R. Vieja | 0.73 (88) | 0.62 (90) | 0.70 (80) | 0.76 (84) |  |

Patch size and rainfall were not correlated (Spearman $= 0.70$, $p = 0.180$), but both factors affected similarly the nested pattern at three different levels: bird community (entire matrix), bird assemblages, and species occurrence among dry forest patches. The value of NODF (overlap and decreasing fill statistics) indicated that the community was significantly nested by size of dry forest patches (NODF $= 31.6$, $p = 0.001$). Furthermore, species occupancy was nested among patches (NODF columns $= 31.6$, $p = 0.001$), but bird assemblages were not nested by size of patches (NODF rows $= 65.6$, $p = 0.099$), though this probability may imply nestedness, with the possibility of an outlier. The entire community also nested within forests ranked by length of dry season (NODF $= 31.6$, $p = 0.001$). The bird assemblages showed a weak tendency to be nested in patches ordered along this gradient of rainfall seasonality (NODF rows $= 32.0$, $0.05 < p < 0.1$) and species occupancy was strongly nested along such gradient (NODF columns $= 31.6$, $p = 0.001$). The similarity of these results indicate that both factors affected the nestedness of species in dry forest patches (Fig. 2), but the small sample size ($N =$ five patches) prevented us from testing the interaction of both factors on species nestedness.

The subset of forest dependent species (categories 1 and 2) was nested when considering area (NODF $= 24.17$, $p = 0.001$) or seasonality (NODF $= 24.15$, $p = 0.001$) of dry forest fragments. Bird assemblages of forest dependent species nested among forest fragments ranked along a seasonality gradient (NODF rows $= 42.15$, $p = 0.005$), but not by area of fragments (NODF rows $= 57.68$, $p = 0.397$). The species occupancy nested along both gradients: seasonality (NODF columns $= 24.13$, $p = 0.001$) and area (NODF columns $= 24.13$, $p = 0.001$). According to Lomolino's test, reduction in species richness for the entire community was not due to habitat loss ($D = 121$, $R = 127.9$, $p = 0.386$, %PN $=$ 6.9), distance between fragments ($D = 120$, $R = 128.9$, $p = 0.336$, %PN $= 6.9$), nor rainfall seasonality ($D = 118$, $R = 128.7$, $p = 0.276$, %PN $= 8.3$). Results are similar for forest dependent species (Table S1).

In general the proportion of species included in the three categories of forest dependency was similar for all sites ($x^2 = 7.7$, $df = 8$, $p = 0.468$; Table 3). The number of forest dependent species (category 1) did not differ across sites ($x^2 = 5.1$, $df = 4$, $p = 0.167$; Table 3). From this category 18 species were detected in only one site and 72% of them were exclusively detected in RV (Table 3). Similarly, for species in category 2 we detected 58% only in RV. From category 3 only four species were detected in only one site.

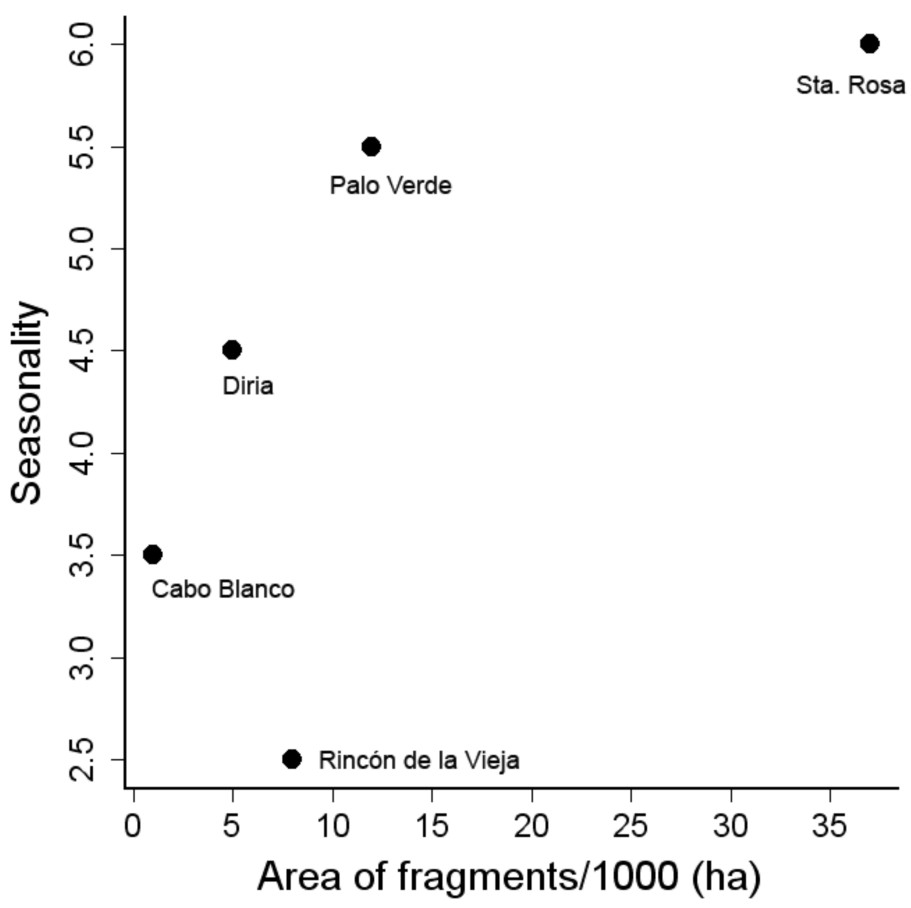

**Figure 2** Relationship between area of fragments and seasonality, estimated as the length of dry season in months.

**Table 3** **Species included in each category of forest dependence.** The first row includes the total number of species in each category and the number of species detected in a single sampling site. The other rows include the number of species of that particular category detected in each site and the number of species detected only in that particular site. The percentage of restricted species of each category per site is shown in parentheses.

| Site | Category 1 | | Category 2 | | Category 3 | |
|------|------------|------------|------------|------------|------------|------------|
| | No. species | Restricted | No. species | Restricted | No. species | Restricted |
| Total | 33 | 18 | 101 | 26 | 53 | 4 |
| Santa Rosa | 10 | 0 (0) | 71 | 1 (4) | 42 | 0 (0) |
| Palo Verde | 14 | 2 (11) | 74 | 4 (15) | 47 | 1 (25) |
| Diriá | 14 | 3 (17) | 58 | 2 (8) | 37 | 2 (50) |
| C. Blanco | 10 | 0 (0) | 61 | 4 (15) | 33 | 0 (0) |
| R. Vieja | 22 | 13 (72) | 70 | 15 (58) | 35 | 1 (25) |

## DISCUSSION

Area did not explain the number of species found in these forest patches, which suggests that other factors, such as environmental heterogeneity, influenced the number and/or composition of species in fragments. For instance, patches with complex topography (e.g., altitudinal gradient, mountains) often have higher species richness, or at least a different subset of species when compared with patches with relatively homogeneous topographies (*Primack, 1998*; *Fernández-Juricic, 2000*; *Mayr & Diamond, 2001*). The characteristics of the matrix surrounding fragments, connectivity between fragments, edge effect, and human interventions may also influence species richness within fragments (*Bierregaard & Stouffer, 1997*; *Whitmore, 1997*).

In this study, nestedness of the dry forest avifauna and forest dependent species is influenced by a gradient of seasonality (length of dry season) and by fragment area (Fig. 2). Sites with a shorter dry season maintain a larger number of dry forest and forest dependent species. From a conservation perspective, forest dependent species are likely more susceptible to global climatic changes, particularly to the changes expected to occur as a consequence of the increasing frequency of ENSO events (*Cai et al., 2014*) and the predicted intensification in the severity of droughts in the region (*Sheffield & Wood, 2008*). Dry forest dependent species require large mature forest tracts to maintain reproductive populations (*Stiles, 1985*). However, both factors, ENSO events and severe droughts, result in longer and more severe dry seasons which consequently increase the frequency of wild fires (*Janzen, 1986*), affecting the physiognomy of the dry forest, changing its composition and structure (*Barlow & Peres, 2004*), and thus affecting the avifauna associated to mature forest tracts.

Fragmentation did not appear to cause a reduction in species richness (based on Lomolinos' test) among forest patches. In systems in which nestedness is caused by the interaction of different factors, as seems to be the case here Lomolinos' test is likely to fail to detect the causes of nestedness. This test has several assumptions (e.g., correlation between nestedness and area of fragments, extinctions are not correlated with isolation) and incorrect assumptions may influence its sensitivity (*Lomolino, 1996*). In this study several factors may obscure the signature of habitat fragmentation (or distance between fragments) on nestedness. For instance, the high resilience apparently inherent to dry forest birds (*Barrantes & Sánchez, 2004*). Many dry forest birds are presumably capable of maintaining small reproductive populations in suboptimal habitats (e.g., small patches of secondary vegetation, pastures, *Barrantes & Sánchez, 2004*). Other species (e.g., *Callocitta formosa* and *Campylorhynchus rufinucha*) are capable of moving between distant forest fragments along linear vegetation corridors or flying between isolated trees or bushes (*Harvey et al., 2005*). Thus, habitat use and behavioral features of some dry forest bird species reduce the probability of detecting the proximal causes of nestedness (e.g., habitat reduction and geographical isolation).

Species composition across sites may be more related to vegetation features than to area of fragments or geographic isolation. The area of the patches included in this study does not predict the number of species present in each fragment. For instance, while having similar species richness, SR is nearly three times the area of PV. Likewise, species richness is

similar in CB and Dir, but the area of CB is only one 20% of the area of Dir. SR and PV are primarily deciduous forest and small tracts of evergreen vegetation (*Hartshorn, 1983*) and both sites have similar species compositions. Dir and CB share many species and both have more humid conditions than SR and PV and larger tracts of evergreen forest, as a result of a shorter dry season (*Janzen, 1986*; *Sánchez-Murillo et al., 2013*). In contrast, RV shares fewer species with other dry forest patches, and populations of many of these species are well isolated by topographic barriers from other populations (*Barrantes, 2009*; *Barrantes, Iglesias & Fuchs, 2011*). The topography of RV is more complex and includes an altitudinal gradient covered by forests with different structure (*Janzen, 1986*). The differences in species composition across sites highlight two important aspects: first that species composition should be analyzed at a finer scale taking forest structure and composition into account; and second, that to preserve the rich dry forest avifauna it is necessary to preserve ecosystem diversity, e.g., through habitat restoration and fragments connection.

Results in Table 3 indicate that between 5 and 12% of all native species in dry forest patches require large areas of mature forests and more than 40% of the species in each forest patch require at least 50% of forest cover for feeding and reproduction (*Stiles, 1985*). Hence, forest patches in northwestern Costa Rica support a large number of species that require large tracts of mature tropical dry forests in the most threaten forest ecosystem in Mesoamerica (*Janzen, 1988*). These patches are then an important reservoir for the rich dry forest Mesoamerican avifauna (*Stotz et al., 1996*), including four endemic species to the Pacific slope of Middle America dry forest region (Lesser Ground Cuckoo, Pacific Screech Owl, Long Tailed Manakin, White Throated Magpie-Jay), but habitat destruction, the removal of isolated trees and forest patches reduce connectivity and may drastically reduce the viability of populations in remnant forest fragments. In these isolated small patches genetic variability may decrease rapidly (*Evans & Sheldon, 2008*, but see *Fuchs & Hamrick, 2010*) and the recurrent catastrophic events caused primarily by intentional fires (*Quesada & Stoner, 2004*) seriously threaten the long-term maintenance of bird populations.

In conclusion, forest patches in northwestern Costa Rica are reservoirs of a large portion of bird species of the Pacific slope of Central American dry forests. However, species composition varies widely across fragments possibly as a consequence of differences in vegetation, climatic and topographic conditions. In northwestern Costa Rica, the reduction of the original dry forest into small, isolated patches resulted in a nested pattern of both bird assemblages and species. The lack of connectivity between these fragments and the recurrent intentional fires in the region, and the predicted global climatic changes threaten the long-term population-viability of many bird species. Nestedness analyses proved to be an important tool to evaluate the consequences of habitat fragmentation of natural environments. Most important, this method can be used periodically to evaluate the effect of changes in climate and land use on the avifuna (or other animals) in forest patches.

## ACKNOWLEDGEMENTS

We thank Julio Sánchez and Luis Sandoval for providing the data for Palo Verde and Rincón de la Vieja respectively. We also thank John Blake, and two other anonymous reviewers, for their critical and useful comments.

### Funding
The authors received no funding for this work.

### Competing Interests
The authors declare there are no competing interests.

### Author Contributions
- Gilbert Barrantes conceived and designed the experiments, performed the experiments, analyzed the data, wrote the paper, prepared figures and/or tables, reviewed drafts of the paper, bird surveys.
- Diego Ocampo and José D. Ramírez-Fernández conceived and designed the experiments, performed the experiments, prepared figures and/or tables, reviewed drafts of the paper, bird surveys.
- Eric J. Fuchs analyzed the data, prepared figures and/or tables, reviewed drafts of the paper.

### Data Availability
The raw data has been supplied as a Supplemental Dataset.

### Supplemental Information
Supplemental information for this article can be found online at http://dx.doi.org/10.7717/peerj.2422#supplemental-information.

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
