# Peer review of "Effect of fragmentation on the Costa Rican dry forest avifauna"

_PeerJ, doi:10.7717/peerj.2422_

## Round 0.1 · original submission · Major Revisions

Dear Authors,

I have also read your manuscript and my comments can be found in the attached manuscript file. Some of my comments are about writing style and others more about the content. The reviewers have noted some details that need to be addressed and I have noted several others in the annotated file. Here I would like to add that I think your map and text should include some reference to, or explanation of, the matrix in which each of your plots is found. Clearly one has ocean around at least half, and so that makes a big difference (or could), and so that context should be more explicit for all. Also, I thought about the possibility of species accumulation curves (by effort). I know you attempt to have a comprehensive list, but an accumulation curve could help. Finally, at least there should be some comparison showing how many species each fragment had in common with each other fragment. I know you have the index of similarity, but, numbers would be nice instead of proportions. I hope you find my comments, and those of the reviewers, very useful.

·

Basic reporting

No comments

Experimental design

No comments

Validity of the findings

No comments

Additional comments

This is a clearly written and scientifically sound paper that provides clear results. The methods were easy to follow and seem appropriate to the questions being asked. In the supplemental table, I would like to see the species identified by category (e.g., forest dependent, etc.). In table 1, can the length of the dry season be given in months – that would provide more information than simply ranking them and would not take up more space; precipitation by season (dry and wet) also would be useful. I realize it is not necessary but I think it might be appropriate to include some photographs that provide perspective on the type of forest and other habitats present in the different fragments; perhaps a google earth shot that indicates the boundaries of the study sites also would be useful – this might provide some perspective on the continuity of the forest, the surrounding matrices and illustrate the connection (or lack thereof) among sites. In line 241 you suggest that many species need at least 50% forest cover to maintain viable populations but do you really have data that addresses viability? Simple presence/absence does not indicate whether the population is viable over the long-term or not, does it? Finally, in lines 253-254 you mention that these fragments are reservoirs for a large portion of dry forest bird species – do you have any estimate of how many (if any) species originally found in these dry forests have already been extirpated?

Reviewer 2 ·

Basic reporting

The introduction, particularly the first paragraph, is very weak and needs to be substantially strengthened and linked to the question of interest. As presented now, it deals with a number of issues and potential hypotheses which the authors are not testing at all.

Experimental design

The research question and knowledge gap need to be more clearly defined. In the discussion, the authors need to elaborate on how their question(s) and results fill this knowledge gap.

Validity of the findings

The science appears to be sound and the findings valid. However, I have made a few suggestions on improving analyses.

Additional comments

L23: I find the use of both “main” and “proximate” redundant. Both essentially mean the same thing. Please use either one of the two words or replace with a word such as “primary”.

L25-26: This sentence is a generalization. Small fragments may not necessarily be poorly interconnected. Neither do small fragments always have to be distantly connected to continuous forests. Please reword appropriately.

L27-33: The authors mention a number of negative effects of fragmentation but do not provide a single link as to how all of this relates to their primary question. All of these make the reader think that the authors are actually going to test hypotheses related to these statements but it is not the case. The introductory paragraph needs major revision in order to flow well and link to the question that the authors are trying to answer.

L46: Provide one or more citations corroborating this statement.

L51: Replace widely spread with widespread. Remove the word interconnected. The word widespread is sufficient to convey the point.

L57-58: The primary question seems to be related to assessing the importance of fragments for dry forest avifauna. The introductory paragraph needs to be completely reworded (see comments above) to link properly with this question.

L59-50: At this stage, it is unclear what the authors mean by “special attention to forest-dependent species......”. Do they only perform analyses for these species? If so, then why mention that they are analyzing data for the bird community as a whole. This sentence needs clarification and rewriting. Perhaps there is more information provided later, but it needs to be very clear what exactly the authors are analyzing.

L61-62: What is the purpose of including a seasonality gradient for this analyses? There is no rationale provided for this so far, only for fragmentation? Please provide an appropriate biological rationale.

L67: Differential extinction – the definition provided reads as a definition for reduction in species richness not extinction. What exactly is this term? Please clarify and provide a citation.

L81: Between years...... What does this mean? Please provide a proper timeline for the study i.e. the year(s) the data was actually collected, any missing data/observations, etc.

L119: A couple of sentences describing the NODF metric would be very helpful for the reader.

L132-134: Reword as: Number of species that depart from perfect nestedness.

L128-140: Is there a more recent analytical method in the community nestedness literature? Why did the authors use Lomolino analysis? There should at least be a rationale for why this particular analysis was selected.

L154: How are the authors inferring species occupancy? Has detection probability been accounted for? It is not clear at all what they mean by occupancy here. I am aware that there is a sentence about this i the introduction. But it deserves proper explanation.

L155: The sentence - .....but bird assemblages did not nest by area among sites – is confusing. The authors mean that the assemblages were not nested. But the sentence reads as if assemblages were not making nests.... Please reword.

L157: It is not advisable to state that a result is “nearly significant”. It is either significant or it is not. Here p is > 0.05 but < 0.1. Perhaps it would be better to use the term “weak tendency”. E.g. “Bird assemblages showed a weak tendency to be nested in fragments ordered...........”.

L171-172: Please state these results explicitly i.e. provide test statistics, etc. Otherwise, the reader just has to assume that what the authors are stating is true.

L186-197: First paragraph of discussion gets too much in the weeds about earlier work and hypotheses that the authors have not tested.

L200: Dependent is spelt incorrectly as dependant. I noticed one or two more such incorrect spellings in other sections of the manuscript as well. Please correct throughout.

L224: I suggest toning down this sentence. The authors have not incorporated any vegetation covariates into their analyses. Therefore, this is speculation and should be written as such. For instance, “Species composition across sites may be related to vegetation........”.

L241: Where is this result coming from? Please explain.

Table 1: In the caption, please provide what exactly the dry season lengths are i.e. how many months does 1 equate to, etc. On what basis, was the length of the dry season classified into these categories. Please provide a rationale.

Table 2: In the caption, provide information on what the Sorensen similarity index numbers mean i.e. higher vs lower numbers – does a higher number mean lower similarity? It will be helpful for the reader.

Reviewer 3 ·

Basic reporting

Quite good, but see the specific comments below. Some additional clarity and focus on the main objectives throughout would be helpful.

Experimental design

Additional detail on the sampling design would be very helpful to evaluate the manuscript fully. In particular greater detail should be provided on the surrounding landscape context, correlations, and how the data were used.

Validity of the findings

Validity of the findings depends on the additional info provided about the experimental design above. So many additional factors could differ between these 5 sites other than those of interest.

Additional comments

This article examines the effect of forest patch size and length of dry season on bird communities in Costa Rica. They examine 5 forest patches (fragments) that vary in size and length of dry season. All of these fragments are quite large remnants. I have number of concerns about the current manuscript that the authors should address.
Major concerns:
7. The small sample size (5) leads to questions of the power to detect effects and how the data were used in the analyses to avoid pseudoreplication. I know the authors detect some effects so if there could be some additional clarity provided regarding the sample units that would be greatly appreciated. Examining power to detect effects would also be great for when differences were not detected (e.g., for area and distance).
8. The use of the terms “area” and “fragmentation is confusing. If the authors are interested in fragmentation they should control for effects of amount of forest surrounding these fragments. It is currently unclear how the surrounding landscape context could affect the results. The authors should read Fahrig 2003 Ann. Rev. Ecol. Evol. Syst. and Fahrig 2013 Journal of Biogeography for why this is relevant here.
9. Given that habitat loss (at a landscape scale) is typically considered to be a major driver of biodiversity declines it I feel that there should be more information on the landscapes surrounding these fragments. Are some of them surrounded by more forest than others etc? How does this affect the results?
10. Given the small sample and the number of variables of interest (i.e., patch size, distance, vegetation structure, seasonality etc.) I would like to know how well interspersed these treatments are. A correlation matrix would be helpful.
11. There could be some improved clarity in the goals and overall synthesis. Sometimes I found it tricky to follow the main objectives.
12. Some of the references are a bit dated and could be supplemented with recent literature.

Specific comments:
Abstract
L 8-11 – I find the extra information in the parentheses to be a bit confusing at this point. Perhaps remove or consider including more detail?
L11 – Area of the forest patch (fragment) or area of forest surrounding the sample site (this is the primary way for area effects to be considered)? Consider using “patch size” instead of area to avoid this confusion
L18 – I suggest including a sentence that synthesizes what these findings mean
Introduction
L25 – Consider newer a citation as well as these?
L27 – Do you mean to say “edge effects” instead of fragmentation here? Might as well be specific. This sounds like you are talking about edge effects, but then there is a separate mention of edge effects later.
L29 – Do you mean
L45 – I would remove “expected” and say something like “have been hypothesized” instead.
L59 – Do you actually mean fragmentation here? In order to examine fragmentation you need to control for surrounding habitat amount while looking at fragmentation. (see Fahrig 2003)
L69 – Habitat availability or patch size?
L90 – I think it would be clearer to call this patch size not area.
L77 – This is not many fragments to be able to separate multiple factors.
L128- How can you separate these two with an N (sample size) of 5??
L150 – How correlated are area and rainfall?
L186 – I would look at Fahrig 2013 Journal of Biogeography

---

## Round 0.2 · Minor Revisions

One of the original reviewers had a few suggestions and comments that I think are valid. In my reviewed copy of your manuscript, I also include some relatively minor suggestions and observations. I find some of the comments about nestedness confusing, because you seem to be saying that a random (non-nested) distribution is expected, but that is not the case. To me, you should emphasize the "nestedness" that explains more of the variation (by rainfall, dry-season length, etc. - similar to the idea of the best r-squared value) rather than simply talking about significant nestedness.

Reviewer 2 ·

Basic reporting

The introduction is better, although I have suggested several changes.

Experimental design

The revised version is much improved in terms of communicating the experimental design.

Validity of the findings

The science is sound and the findings valid.

Additional comments

L38: Habitat degradation would perhaps be more suitable than “deterioration”

L40-41: Reword as “These changes in natural landscapes may often reduce connectivity for species trying to move between fragments embedded in a matrix that consists of anthropogenic and semi-natural habitats.”

L42: Habitat loss and fragmentation are not the same thing. Refer to Fahrig 2003. Annual Review of Ecology, Evolution and Systematics. In this paper, the authors appear to be testing the effects of habitat fragmentation and not habitat loss. Therefore, I would refrain from using the term habitat loss.

L42-43: Reword as “Additionally, the negative effects of habitat fragmentation may be exacerbated by increased predation and competition within habitat patches.

L44: Again, it would be advisable to not mix habitat loss and fragmentation.

L49: The term species richness cannot be linked to a species. Richness is associated with communities. The sentence could be reworded as “………fragmentation could increase bird species richness and the abundance of species………”

L50-51: What is the difference between richness and diversity? The authors should first define a term (e.g. richness or diversity) on the basis of the question of interest and then stick to it throughout.

L70: Are you testing for the effect of loss or for the effect of fragmentation?

L80: What do the authors mean by “……the likely cause of reduction of species…..” What is reduction exactly – lower species richness, extinction?

L187: Differential extinction needs to be explained or removed. It has been removed from the Introduction but also needs to be removed here. Same for L218. Either explain or remove.

---

## Round 0.3 · accepted · Accept

I am enclosing your final manuscript with a few, very minor, suggestions and corrections to improve the text a bit. Nothing major, nothing conceptual. Also, I'd recommend you look over your use of the words "significant" and "significantly." In my mind, once you say things are different, they should be statistically different and the use of the word significant is superfluous. So, if used correctly, the words SHOULD indicate conceptual IMPORTANCE rather than statistical significance.

These are minor issues and you can fix them with the production group.